# Non-Solvent Synthesis of a Robust Potassium-Doped PdCu-Pd-Cu@C Nanocatalyst for High Selectively Tandem Reactions

**Sanha Jang** [1,†], **Dicky Annas** [1,†], **Sehwan Song** [2], **Jong-Seong Bae** [3], **Sungkyun Park** [2,*] and **Kang Hyun Park** [1,*]

[1]  Department of Chemistry, Pusan National University, Busandaehak-ro 63beon-gil, Geumjeong-gu, Busan 46241, Korea; jangs0522@naver.com (S.J.); annas.dickyy@gmail.com (D.A.)
[2]  Department of Physics, Pusan National University, Busandaehak-ro 63beon-gil, Geumjeong-gu, Busan 46241, Korea; sehwan465@gmail.com
[3]  Busan Ceter, Korea Basic Science Institute, Gwahaksandan 1-ro 60beon-gil, Gangseo-gu, Busan 46742, Korea; jsbae@kbsi.re.kr
*  Correspondence: psk@pusan.ac.kr (S.P.); chemistry@pusan.ac.kr (K.H.P.); Tel.: +82-51-510-2595 (S.P.); +82-51-510-2238 (K.H.P.)
†  These authors contributed equally to this work.

**Abstract:** A non-solvent synthesis of alkali metal-doped PdCu-Pd-Cu@C is presented that needs no mechanical grinding and utilizes heat treatment under an $N_2$ gas flow. Pluronic® F127 is used to generate pores and a high surface area, and tannic acid is used as a carbon source for the PdCu-Pd-Cu@C nanocatalysts. Because some C is transferred to organic compounds during the nitrogen heat treatment, this demonstrated the advantage of raising the weight ratio of active metals comparatively. The PdCu-Pd-Cu@C nanocatalyst developed in this study outperformed commercial Pd/C catalysts by bimetallic PdCu-Pd-Cu nanoparticles and Pd nanoparticles in terms of catalytic activity (selectivity of commercial Pd/C: 45%; PdCu-Pd-Cu@C nanocatalyst: 76%). The alkali metal dopants increase the selectivity of the final product on the PdCu-Pd-Cu@C surface because they are electron-rich, which assists in the adsorption of the substrate (selectivity of PdCu-Pd-Cu@C nanocatalyst: 76%; K-doped PdCu-Pd-Cu@C nanocatalysts: 90%). Furthermore, even after being reused 5 times in this research, the final catalytic performance was comparable to that of the initial catalyst.

**Keywords:** non-solvent synthesis; PdCu alloy; bimetallic nanocatalyst; tandem reaction; promoter; alkali metal doping

## 1. Introduction

In recent decades, hybrid multimetallic nanoparticles (NPs) have been utilized as catalysts, due to their high catalytic performance and physico-chemical stability relative to comparable metal-based catalysts [1–11]. Pd and Cu nanomaterials have generated increased interest owing to the high catalytic activity (Pd) and synergistic impact (Cu) of each nanocatalyst component. Our previous study presented a hybrid nanocatalyst based on Cu-doped Pd-Fe$_3$O$_4$ [12], but the use of sodium oleate and 1-octadecene as solvents for the synthesis of the nanocatalyst may present human health and ecotoxicity concerns.

The typical processes used for the synthesis of NPs include co-precipitation, electrosynthesis, wet chemical reduction, microemulsion, and hydrothermal synthesis [13–21]. Alternatively, eco-friendly syntheses utilize a solvent-free process, such as grinding or melt infiltration [22–29]. In particular, grinding via ball-milling is a highly efficient and environmentally sustainable approach to the manufacture of nanomaterials [22]. Gao et al. proposed ball-milled carbon nanomaterials for energy and environmental applications [23] and recently published more details about the ball-milling process. This technique has gained popularity and was successfully applied to the manufacture of a wide range of carbon nanomaterials and their derivatives with improved physico-chemical properties.

Dai et al. reported the solid-state synthesis of a catalyst composed of ordered mesoporous carbons (OMCs) using a grinding process [30]. They demonstrated a flexible mechanochemical self-assembly approach to the manufacture of metal polyphenols ($Zn^{2+}$ or $Ni^{2+}$) and poly(ethylene oxide)–poly(propylene oxide)–poly(ethylene oxide) mesophase composites prepared by ball-milling that allow for the simple synthesis of ordered mesoporous carbons after the pyrolysis procedure.

The use of a promoter is one method to improve the function of a catalyst [31–36]. In particular, the use of a small amount of alkali metal as a promoter can effectively boost catalytic efficiency. However, at high concentrations of alkali metals, the efficiency of the catalyst decreases [37]. Park and coworkers studied K-doping, using computational chemistry and Fischer–Tropsch experiments focused on K-doped iron carbide, and they found similar results when carbon and aluminum were used as support materials [27,38]. The same effect has also been observed for cesium (Cs), another alkali metal [39]. Kotarba et al. reported an explanation of the improvement in the electronic properties of $Co_3O_4$ after doping with K. The experimental data demonstrated work function improvements after K-doping and were supported by density functional theory (DFT) calculations [37,38].

Benzofuran is a heterocyclic compound that is commonly used in pharmaceuticals because of its biological activity. Many drugs contain benzofuran or its derivatives, which are bound to other cyclic or heterocyclic compounds [40]. Several studies have reported that benzofuran derivatives demonstrate bioactivity as antifungal, antimicrobial, and antitumor agents [41–43]. The synthesis of benzofuran derivative compounds can be performed using a variety of methods. One method includes using alkynyl carboxylic acids as a precursor because of their high reactivity and availability. Woo et al. developed a benzofuran derivative via a tandem reaction method using 2-iodophenol and phenylpropiolic acid as precursors and Cu-doped Pd-$Fe_3O_4$ NPs as the catalyst to form 2-phenylbenzofuran with a high conversion [12].

Herein, the synthesis of PdCu-Pd-Cu@C and K-doped PdCu-Pd-Cu@C nanocatalysts based on a non-solvent method of synthesis is reported, followed by an application of tandem reactions (Scheme 1). Furthermore, the synergistic effect of PdCu-Pd-Cu was demonstrated by comparing doping with other alkali metals, such as Cs and Na, and commercial Pd/C, $Cu_2O$, and CuO. These K-doped PdCu-Pd-Cu nanocatalysts exhibited high catalytic performance and recyclability for the tandem reaction of 2-phenylbenzofuran (Conv.: >99% and Sel.: 90%). Moreover, even after 5 reuses to evaluate the catalyst life, the performance of the catalyst may be maintained at up to 87%.

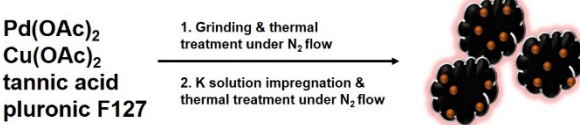

**Scheme 1.** Preparation of K-doped PdCu-Pd-Cu@C nanocatalyst via heat treatment under $N_2$ gas flow.

## 2. Results and Discussion

### 2.1. Preparation of K-Doped PdCu-Pd-Cu @C Nanocatalyst

The synthesis of K-doped PdCu-Pd-Cu@C NPs has been previously achieved by mechanical grinding and wetness impregnation [27,30,38,39]. In the current study, when the heat treatment is carried out at 450 °C under a flow of $N_2$, mesopores in the nanocatalyst are generated when the Pluronic® F127 decomposes, and tiny PdCu-Pd-Cu NPs are formed in the mesoporous carbon. After thermal treatment and submersion in ethanol, the PdCu-Pd-Cu@C powder was fully dried in a vacuum oven at 60 °C. The K-doped PdCu-Pd-Cu@C nanocatalyst was collected using a wetness impregnation process with an aqueous $K_2CO_3$ solution as the K source.

The low-resolution transmission electron microscopy (TEM) images show black dots that indicate incorporated NPs of PdCu-Pd-Cu and K-doped PdCu-Pd-Cu nanocatalysts

(Figure 1a,b). For the PdCu-Pd-Cu@C nanocatalyst, the average PdCu-Pd-Cu diameter was about 7.2 nm, and the average diameter for the K-doped PdCu-Pd-Cu@C nanocatalysts increased marginally in size to about 11.9 nm (inset of Figure 1a,b). A high-resolution transmission electron microscopy (HRTEM) of K-doped PdCu-Pd-Cu@C nanocatalyst was performed to study the lattice of single Pd, PdCu, and Cu NPs embedded in the carbon material (Figure 1c,d). The Fourier transform pattern indicated a single crystal of metallic Pd with a distance of 0.23 nm between the adjacent fringes, corresponding to the (111) plane of the centered cubic Pd face. Lattices of PdCu were found with a distance of 0.22 nm between the adjacent fringes, corresponding to the (111) plane. In addition, lattices of Cu were found with a distance of 0.21 nm between the adjacent fringes, corresponding to the (111) plane of the centered cubic Cu face.

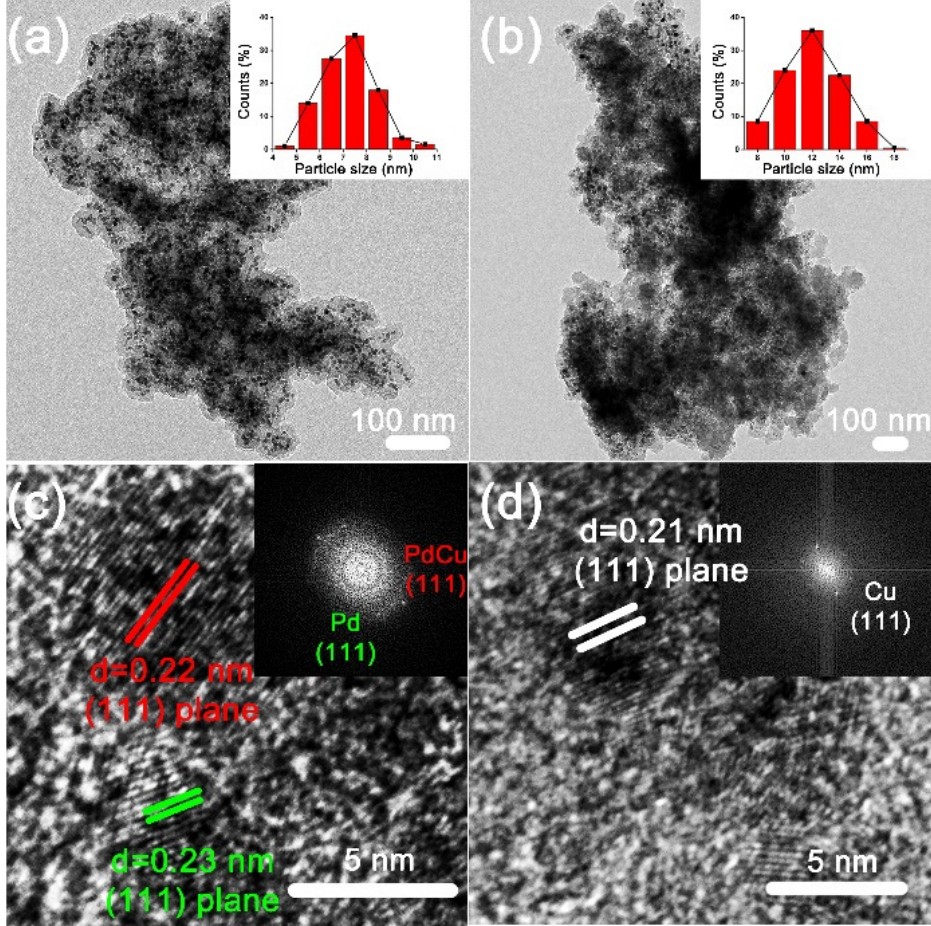

**Figure 1.** TEM images of (**a**) PdCu-Pd-Cu@C nanocatalyst and (**b**) K-doped PdCu-Pd-Cu@C (inset of PdCu-Pd-Cu NPs size distribution). K-doped PdCu-Pd-Cu@C nanocatalyst of (**c,d**) HRTEM image (inset of FT patterns).

The high-angle annular dark-field scanning transmission electron microscopy (HAADF-STEM) images of K-doped PdCu-Pd-Cu@C nanocatalyst exhibited tiny bright dots, indicating the uniform penetration of the PdCu-Pd-Cu NPs into the mesoporous carbon (Figure 2a). The elemental mapping found Cu (blue), Pd (green), C (red), and K (cyan) and indicated a high dispersion of Cu and Pd (Figure 2b–d). K was also found to be well-dispersed on the nanocatalyst. In addition, the elemental mapping of the PdCu-Pd-Cu@C nanocatalyst (Figure S1) indicated C (red), Cu (blue), and Pd (green).

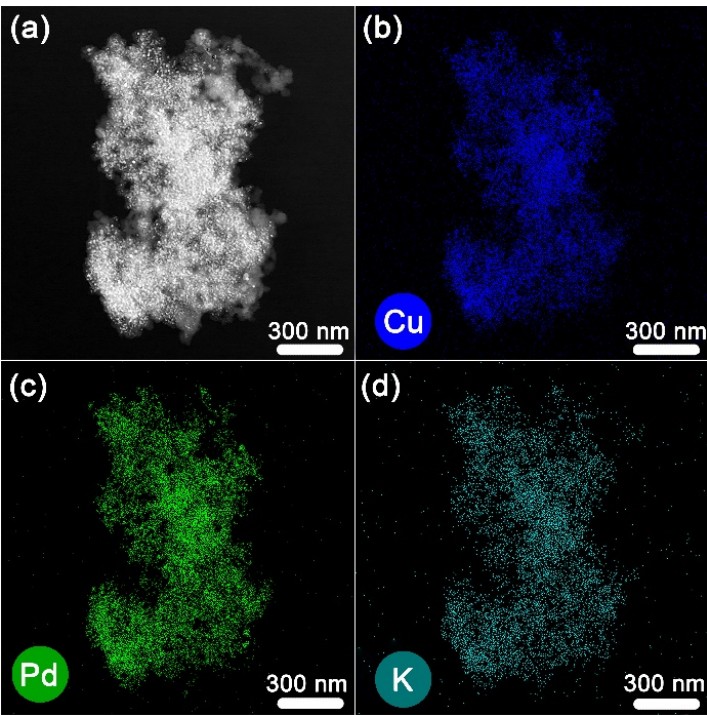

**Figure 2.** K-doped PdCu-Pd-Cu@C nanocatalyst of (**a**) HAADF image and (**b**–**d**) elemental mapping images. (**b**) Cu (blue), (**c**) Pd (green), and (**d**) K (cyan), respectively.

In the XRD patterns of PdCu-Pd-Cu@C and K-doped PdCu-Pd-Cu@C powders, Pd peaks were observed at 2θ values of 40.0°, 46.5°, and 69.9°, corresponding to reflections of the (111), (200), and (220) planes of fcc Pd, respectively (Figure 3a, JCPDS No. 88-2335). Cu peaks were observed at 2θ values of 43.3°, 50.4°, and 74.1°, corresponding to reflections of the (111), (200), and (220) planes of fcc Cu, respectively (JCPDS No. 85-1326). In addition, PdCu alloy peaks were observed at 2θ values of 41.4°, 48.2°, and 70.5°, corresponding to reflections of the (111), (200), and (220) planes of fcc PdCu, respectively (JCPDS No. 48-1551). The spectra of the K-doped PdCu-Pd-Cu@C nanocatalysts exhibited additional $Cu_2O$ peaks at 2θ values of 29.6°, 36.4°, and 42.3°, corresponding to reflections of the (110), (111), and (200) planes of $Cu_2O$, respectively (JCPDS No. 78-2076). The $Cu_2O$ peaks in the K-doped PdCu-Pd-Cu@C nanocatalyst exhibit more crystallinity than the PdCu-Pd-Cu@C peak due to sintering that occurred during reheat treatment, so the particle size of $Cu_2O$ increased.

The composition and chemical state of the PdCu-Pd-Cu@C and K-doped PdCu-Pd-Cu@C nanocatalysts were confirmed by XPS. Survey scans indicated that both samples contain Pd, Cu, and O without any other contaminating elements (Figure S2a). The spectra were analyzed after calibration using C–C bonds in the C 1*s* spectra (Figure S2b). The binding energy of C–C (284.5 eV) in the C 1*s* spectra was determined after deconvoluting the spectra (Figure S2b) into four different chemical states, including C–C, C–O–C, O–C=O, and COOH, and a shake-up feature induced by a π–π* transition [44]. Figure 3b shows the doublet state of Pd $3d_{5/2}$ (335.10 eV) and Pd $3d_{3/2}$ (340.36 eV) caused by spin-orbit splitting, indicating that the Pd is in a metallic state (e.g., Cu–Pd and Pd–Pd) [45]. Figure 3c also shows the Cu 2*p* spectra for both samples. However, unlike in the Pd spectra, the Cu spectra indicated two different chemical states at 931.80 eV for Pd–Cu and 933.33 eV for $Cu_2O$ (or Cu–Cu) [46]. Furthermore, the more detailed narrow scan shows the metallic K 2*s* state (Figure 3d) in the K-doped PdCu-Pd-Cu@C. This result indicated the presence of metallic potassium on the catalyst surface, which was not visible in XRD.

FT-IR spectra of PdCu-Pd-Cu@C and K-doped PdCu-Pd-Cu@C nanocatalyst are showed in Figure 4a. The band at 1740 cm$^{-1}$ is the carboxylate that defined the asymmetric stretching vibration of C=O. The presence of C=C stretching was suggested by

the appearance of a strong signal at 1576 cm$^{-1}$. The strong peaks at 1166 and 1043 cm$^{-1}$ were assigned to C-O stretching. C-O stretching showed in the strong peaks at 1166 and 1043 cm$^{-1}$. Even after heat treatment, the PdCu-Pd-Cu@C and K-doped PdCu-Pd-Cu@C nanocatalysts exhibited no changes in the peak, and the presence of K was unknown, but it was closely connected to the carbon analysis in Figure S4b.

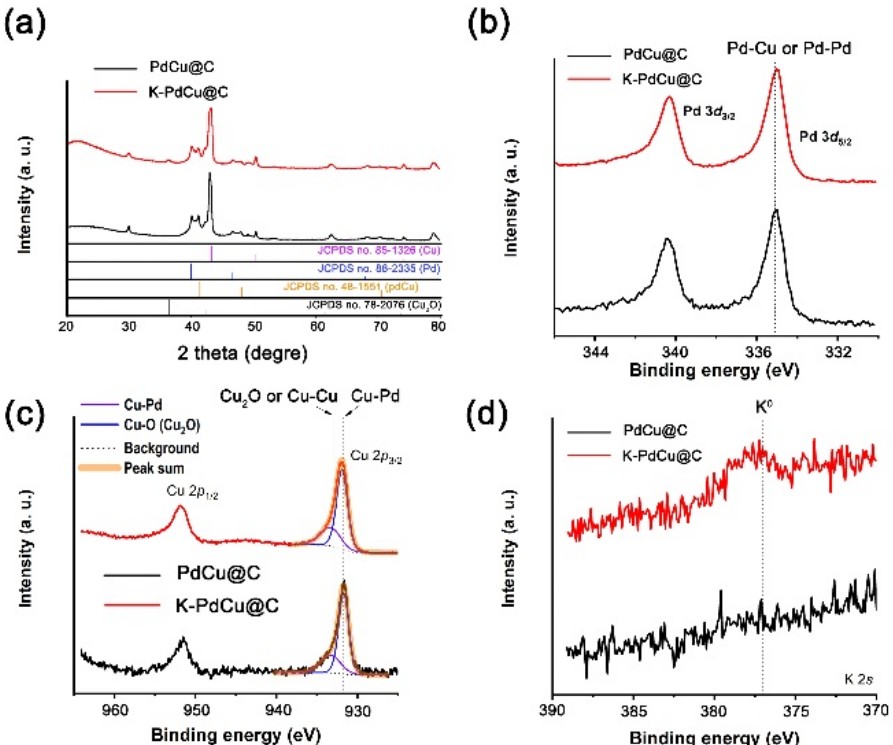

**Figure 3.** (**a**) XRD spectra of the PdCu-Pd-Cu@C and K-doped PdCu-Pd-Cu@C nanocatalysts. (**b**–**d**) XPS spectra of PdCu-Pd-Cu@C and K-doped PdCu-Pd-Cu@C nanocatalysts.

TGA analysis is a technique to determine thermal stability, and Figure 4b shows the thermal stability of the PdCu-Pd-Cu@C and K-doped PdCu-Pd-Cu@C nanocatalysts. Due to loss of solvent or water in the catalyst, the weight of materials decreased when the temperature reached 100 °C. Then, the weight of materials increased due to the phase transition of Pd and Cu in the catalyst, verifying the relative change by normalization. Furthermore, both catalysts experience weight loss after reaching 350 °C. Both catalysts dropped in weight quickly after reaching 350 °C. Although the weight changed, the K-doped PdCu-Pd-Cu@C nanocatalyst weight change was smaller than PdCu-Pd-Cu@C. As a result of their excellent thermal stability, they can be applied in tandem processes.

N$_2$ sorption experiments were performed on PdCu-Pd-Cu@C and K-doped PdCu-Pd-Cu@C nanocatalysts and demonstrated a type IV adsorption–desorption hysteresis (Figure 4c). The Brunauer–Emmett–Teller (BET) surface area and total pore volume were calculated to be 270.51 m$^2$·g$^{-1}$ and 0.163 cm$^3$·g$^{-1}$ for the PdCu-Pd-Cu@C nanocatalyst, respectively, and 270.40 m$^2$·g$^{-1}$ and 0.185 cm$^3$·g$^{-1}$ for the K-doped PdCu-Pd-Cu@C nanocatalyst, respectively. The pore sizes of the PdCu-Pd-Cu@C and K-doped PdCu-Pd-Cu@C nanocatalysts were measured at 11.10 nm and 12.55 nm, respectively, based on the desorption branch using the Barrett–Joyner–Halenda (BJH) method (Figure 4d). Due to the production of CH$_4$, CO$_2$, CO, and organic compounds in tannic acid during the heat treatment, the pore volume and size of the K-doped nanocatalyst are enhanced.

Using an ICP-OES, the Pd loading in the K-doped PdCu-Pd-Cu@C nanocatalyst was found to be 19.9 wt% (Table S1). The theoretical Pd content of the K-doped PdCu-Pd-Cu@C nanocatalyst, based on the Pd(OAc)$_2$ after the heat treatment, was calculated to be 15.1 wt%. Based on these results, it is assumed that the weight of C was lowered because

some components, such as $CH_4$, $CO_2$, CO, and organic compounds, were produced as by-products of the heat treatment of the PdCu-Pd-Cu@C nanocatalyst. As a result, the active metal content of the catalyst increased.

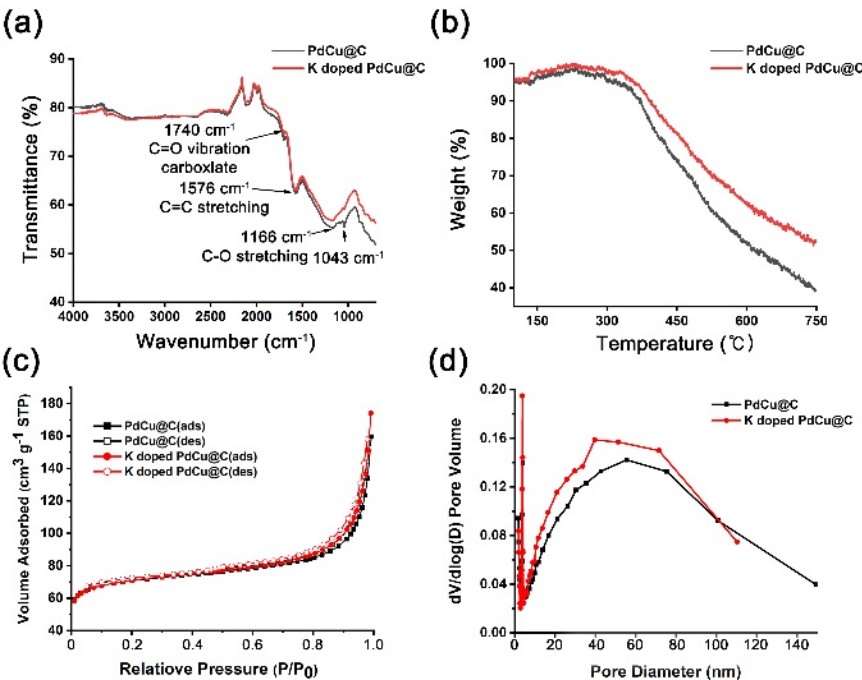

**Figure 4.** FT-IR spectra of (**a**) PdCu-Pd-Cu@C and K-doped PdCu-Pd-Cu@C nanocatalysts. (**b**) TGA curves of the PdCu-Pd-Cu@C and K-doped PdCu-Pd-Cu@C nanocatalysts. (**c**) $N_2$ adsorption/desorption isotherms. (**d**) Pore size distribution diagrams, calculated from desorption branches using the BJH method.

### 2.2. The Catalytic Activity of Nanocatalysts toward the Tandem Reaction of 2-Benzofuran

The catalytic activity of the K-doped PdCu-Pd-Cu@C nanocatalyst was investigated for the tandem reaction of 2-benzofuran, using phenyl propiolic acid and 2-iodophenol under several reaction conditions (Table 1). Initially, 1.3 mol% of the catalyst was used, with NaOAc as the base and DMSO as the solvent, and the reaction was conducted at 130 °C for 1, 3, or 5 h. A temperature of 130 °C was chosen because this was the optimum temperature reported for this kind of reaction in previous work [12]. We analyzed the product using GC-MS and $^1$H NMR for 2-phenylbenzofuran. Our findings are well-matched in terms of mass spectra and $^1$H NMR data (Figures S3 and S4). When the reaction time was 3 and 5 h, the conversion of reactants to the target product was high (more than 90%); however, the selectivity was low (lower than 50%) (Table 1, entries 1 and 2). Furthermore, when the reaction time was 1 h, the conversion decreased to 84%, but the selectivity increased to 55% (Table 1, entry 3). This latter result indicates that this reaction can happen over a short time. We utilized a variety of bases to optimize this reaction. When $Cs_2CO_3$ and $NaHCO_3$ were used as the base, low conversion and selectivity were observed, which indicates that these bases are not suitable for this reaction (Table 1, entries 4 and 5). Similar low reactivity was observed when LiOAc was used as the base, with a conversion and selectivity of 72% and 49%, respectively. Interestingly, when a strong base was used (KOH), the conversion decreased to 73% and the selectivity increased to 87% (Table 1, entry 7). As a comparison, NaOH was also used, but the conversion and selectivity were lower than for KOH. Given that the selectivity is more important for this reaction than the conversion, KOH was selected as the base for this reaction. In an effort to increase the conversion, the amount of catalyst was increased to 4 mol%. The conversion increased to 88% and almost no selectivity difference was observed between 1.3 mol% and 4.0 mol% of the catalyst (Table 1, entry 9). When the amount of catalyst was increased to 6.6 mol%, the highest

conversion and selectivity were observed, >99% and 90%, respectively (Table 1, entry 10). Therefore, 6.6 mol% of catalyst, KOH as the base, DMSO as the solvent, 130 °C, and 1 h of reaction time were selected as the optimal conditions and used for further optimization of the various nanocatalysts.

**Table 1.** Optimization of tandem reaction conditions using K-doped PdCu-Pd-Cu@C nanocatalyst.

| Entry | Amount of catalyst (mol%) | Base | t (h) | Conv. (%) [1] | Sel. (%) |
|-------|---------------------------|------|-------|---------------|----------|
| 1 | 1.3 | NaOAc | 5 | 90 | 43 |
| 2 | 1.3 | NaOAc | 3 | 82 | 43 |
| 3 | 1.3 | NaOAc | 1 | 84 | 55 |
| 4 | 1.3 | $Cs_2CO_3$ | 1 | n.d. | n.d. |
| 5 | 1.3 | $Na_2CO_3$ | 1 | 8 | <10 |
| 6 | 1.3 | LiOAc | 1 | 72 | 49 |
| 7 | 1.3 | KOH | 1 | 73 | 87 |
| 8 | 1.3 | NaOH | 1 | 63 | 71 |
| 9 | 4.0 | KOH | 1 | 88 | 82 |
| 10 | 6.6 | KOH | 1 | >99 | 90 |

Reaction conditions: K-PdCu-Pd-Cu@C (K/Pd molar ratio: 0.06, Pd base: 19.9 wt%, and Cu base: 13.3 wt% by ICP-OES data), 2-iodophenol (0.5 mmol), phenyl propiolic acid (0.6 mmol), base (1.0 mmol), and solvent (5.0 mL). DMSO for solvent and 130 °C for temperature. [1] Determined by using GC-MS spectroscopy based on 2-iodophenol. n.d.: not detected.

As a comparison, the activity of other heterogeneous catalysts was evaluated toward this tandem reaction, using the optimum conditions. First, the activity of commercial Pd/C as a catalyst was measured and demonstrated a high conversion (>99%) but a low selectivity (45%) (Table 2, entry 1). When commercial $Cu_2O$ and CuO were used as catalysts, no product conversion was observed (Table 2, entries 2 and 3), indicating that these catalysts are not strong enough to convert the reactants to product. The PdCu-Pd-Cu@C nanocatalyst was also synthesized as a comparison and demonstrated a high conversion even without K doping, but the selectivity was lower than in the reaction with K-doped PdCu-Pd-Cu (Table 2, entry 4). This result indicates that K activities play an important role in the tandem reaction for selectivity. Furthermore, other alkali metals, such as Na and Cs, were screened as promoters for this reaction. These two catalysts gave a high conversion (>99%), but were still lower selective than the K-doped PdCu-Pd-Cu@C catalyst (Table 2, entries 6 and 7). This indicates that there is a synergistic effect between Pd, Cu, and K. In addition, it is likely that KOH being used as the base also has a synergistic effect with the K in the catalyst, to increase the basicity of the catalyst. After the reaction, the K-doped PdCu-Pd-Cu@C catalyst was isolated and washed several times with water and methanol, and then dried in a vacuum oven.

### 2.3. Recyclability Test

The recyclability of the K-doped PdCu-Pd-Cu@C nanocatalyst for the tandem reaction was investigated under optimum conditions. The catalyst was washed and collected by centrifugation to be reused five times. Figure 5 shows the catalytic activity after these cycles and demonstrates that the catalyst is still active with a >99% level of conversion and 88.4% of average selectivity. These results indicate that the K-doped PdCu-Pd-Cu@C nanocatalyst has an excellent activity toward this tandem reaction.

**Table 2.** Tandem reaction of 2-phenylbenzofuran using various nanocatalysts.

| Entry | Catalyst | Conv. (%) [1] | Sel. (%) |
|-------|----------|---------------|----------|
| 1 | Pd/C | >99 | 45 |
| 2 | $Cu_2O$ | n.d. | n.d. |
| 3 | CuO | n.d. | n.d. |
| 4 | PdCu-Pd-Cu@C | >99 | 76 |
| 5 | K-PdCu-Pd-Cu@C | >99 | 90 |
| 6 | Na-PdCu-Pd-Cu@C | >99 | 82 |
| 7 | Cs-PdCu-Pd-Cu@C | >99 | 78 |

Reaction conditions: 2-iodophenol (0.5 mmol), phenyl propiolic acid (0.6 mmol), base (1.0 mmol), and DMSO (5.0 mL). Settings: 6.6 mol% for the amount of catalyst, KOH for the base, solvent for DMSO, and 130 °C for temperature. [1] Determined by using GC-MS spectroscopy based on 2-iodophenol. n.d.: not detected.

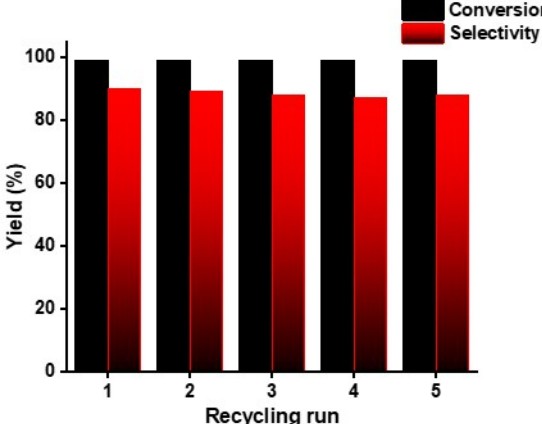

**Figure 5.** The recyclability of K-doped PdCu-Pd-Cu@C for tandem reaction.

Furthermore, to measure the stability of the material after the catalyst, TEM characterization was used (Figure 6). After five cycles, the PdCu-Pd-Cu NP size of the K-doped PdCu-Pd-Cu@C nanocatalyst was maintained and only a small aggregation of particles was observed. Moreover, using ICP-OES, the metal content of the used catalyst was studied. As shown in Table S2, the content of metals increased compared to the unused catalyst. This increase in metal content may be due to the loss of carbon during the washing process.

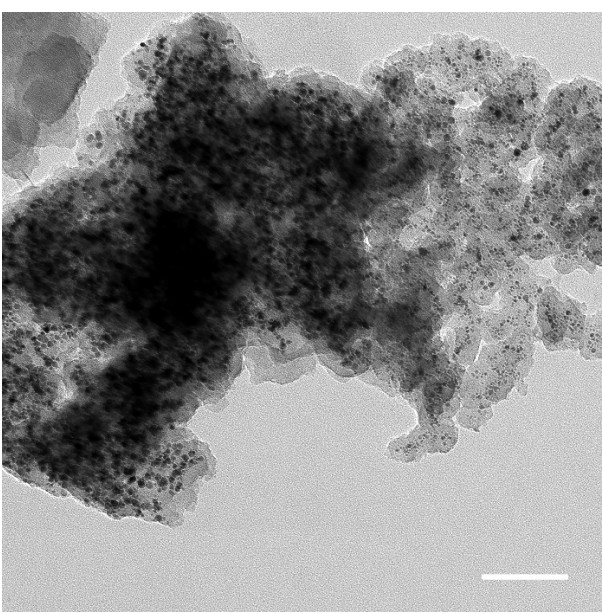

**Figure 6.** TEM image of the recovered K-doped PdCu-Pd-Cu@C nanocatalyst after five cycles of reaction. The bar represents 100 nm.

## 3. Materials and Methods

### 3.1. Chemicals and Characterization

Palladium(II) acetate (Pd(CH$_3$COO)$_2$, 98%), copper(II) acetate (Cu(CH$_3$COO)$_2$, 98%), Pluronic$^®$ F127, 2-iodophenol (C$_6$H$_5$IO, 99%), and phenyl propiolic acid (C$_9$H$_6$O$_2$, 99%) were purchased from Aldrich (St. Louis, MO, USA). Tannic acid (C$_{76}$H$_{52}$O$_{46}$) was purchased from Alfa Aesar. Potassium carbonate (K$_2$CO$_3$, 99.5%), ethanol (C$_2$H$_5$OH, 99.5%), dichloromethane (CH$_2$Cl$_2$, 99.5%), sodium bicarbonate (NaHCO$_3$, 99.0%) and dimethyl sulfoxide ((CH$_3$)$_2$SO, 99.0%) were purchased from Samchun (Pyeongtaek, Korea). Potassium hydroxide (KOH, 85–87%) and magnesium sulfate anhydrous (MgSO$_4$) was purchased from Daejung (Siheung, Korea). All reagents were used as received, without further purification.

The morphology and elemental mapping data of the samples were analyzed using field-emission transmission electron microscopy (FE-TEM, TALOS F200X) (Thermo Scientific, MA, USA) and energy-dispersive X-ray spectroscopy (XPS, K-alpha) (Thermo Scientific, MA, USA). The samples for TEM were prepared by adding a small amount of the colloidal solution to carbon-coated copper grids (Ted Pella, Inc) (CA, USA). The X-ray diffraction (XRD) spectra were recorded on a Rigaku D/MAX-RB diffractometer (Tokyo, Japan). ATR-FTIR spectra were recorded using a Thermo Scientific Nicolet iS10 FT-IR spectrometer (MA, USA). Thermogravimetric analysis (TGA) was performed on a TGA-50-Shimadazu analyzer (Kyoto, Japan) in the range from 25 to 800 °C at a heating rate of 10 °C·min$^{-1}$. Nitrogen sorption isotherms were measured at 77 K using a Tristar II 3020 (GA, USA) surface area analyzer. Before measurement, the samples were degassed under vacuum at 573 K for 4 h. The Pd, Cu, and K loading amounts were measured using an inductively coupled plasma-optical emission spectrometer (ICP-OES, PerkinElmer Optima 8300) (MA, USA). X-ray photoelectron spectroscopy (XPS, K-Alpha XPS system, Thermo Fisher Scientific) was performed to confirm the chemical state and composition using a monochromatic Al K$\alpha$ (hν = 1486.6 eV) at the Busan Center, Korea Basic Science Institute. Gas chromatography–mass spectrometry (GC-MS) was performed on a Shimadzu GC-1010 Plus GCMS-QP2010 SE (Kyoto, Japan).

### 3.2. Synthesis of PdCu-Pd-Cu@C and K-Doped PdCu-Pd-Cu@C Nanocatalysts

Palladium(II) acetate (0.225 g), copper(II) acetate (0.182 g), tannic acid (1 g), and Pluronic$^®$ F127 (0.7 g) were mechanically ground in a mortar for 5 min under atmospheric conditions to produce a uniform light brown powder. After grinding, the powdered mixture was placed in an alumina boat and slowly heated at a ramp rate of 3.5 °C min$^{-1}$ to 450 °C under a flow of N$_2$ gas (0.2 L min$^{-1}$), and then held at 450 °C for 4 h. After the heating process, the resulting black powder was cooled to room temperature (RT) and then immersed in ethanol (35 mL) under a continuous flow of N$_2$ (1 L min$^{-1}$) to mitigate oxidation of the active surface. The powder that was submerged in ethanol was collected using centrifugation and dried in a vacuum oven at 60 °C.

For the K doping, K$_2$CO$_3$ powder (0.049 g) was dissolved in deionized water (10 mL), and then a small amount of the K$_2$CO$_3$ solution (0.1 mL) was added to the as-prepared PdCu-Pd-Cu@C nanocatalyst (0.1 g). The sample with added K was heat-treated again at 350 °C for 4 h under a flow of N$_2$ at 0.2 mL min$^{-1}$. The resulting black powder was then cooled to RT and again immersed in ethanol under a flow of N$_2$ gas. Finally, the K-doped PdCu-Pd-Cu@C nanocatalyst was isolated and dried in a vacuum oven at 60 °C.

### 3.3. Tandem Reaction of 2-Phenylbenzofurans

The tandem reaction from our previous work [12] was followed with minor modifications. The K-doped PdCu-Pd-Cu@C nanocatalyst (molar ratio of K/Pd: 0.06), 2-iodophenol (0.5 mmol), phenyl propiolic acid (0.6 mmol), potassium hydroxide (1.0 mmol), and solvent (5.0 mL) were added to a round-bottom flask and connected to a reflux system. The mixture was stirred at reflux under air for 1 h. After the reaction, the catalyst and product mixture was filtered, and the product was extracted several times using a solution of

dichloromethane and $NaHCO_3$. The product was then dried using $MgSO_4$, filtered, and evaporated using a rotary evaporator. The product was analyzed using GC-MS.

### 3.4. Recyclability Test

To reuse the catalyst, it was separated from the mixture and washed several times with distilled water and MeOH and then collected by centrifugation to be reused for five cycles. Furthermore, following the reaction, the catalytic performance was evaluated using GC-MS.

## 4. Conclusions

A K-doped PdCu-Pd-Cu@C nanocatalyst was conveniently prepared using the non-solvent synthesis of metal acetates and a wetness impregnation with an aqueous K solution. A high metal load (Pd: 19.9 wt%, Cu: 13.3 wt%, and K: 0.47 wt%, a molar ratio of K/Pd = 0.06) and catalytic dispersion led to an increased efficiency and particle stability for tandem reactions. The catalyst exhibited a high catalytic conversion (>99%) under optimized tandem reaction conditions (6.6 mol%) as compared to the well-dispersed PdCu and Pd NPs (11.9 nm). In particular, the K-doped PdCu-Pd-Cu@C nanocatalyst showed a higher selectivity (>88%) than the PdCu-Pd-Cu@C nanocatalyst. It is likely that K encourages strong 2-iodobenzene adsorption on the catalytic surface. Furthermore, recycling tests where the catalyst was reused five times at a temperature of 130 °C demonstrated the same conversion and selectivity as the initial catalytic performance. This synthetic technique for forming bimetallic PdCu-Pd-Cu catalysts could be extended to other metal-noble metal hybrid nanocatalysts, such as mesoporous carbon supported on CoPd, FePd, or NiPd, for future catalyst development. Furthermore, it revealed the capacity to use alkali metal to manufacture catalysts with high performance in the absence of solvents.

**Supplementary Materials:** The following are available online at https://www.mdpi.com/article/10.3390/catal11101191/s1, Figure S1. HAADF-STEM image and elemental mapping images of PdCu@C nanocatalyst, Figure S2. XPS spectra of K doped PdCu@C nanocatalyst, Figure S3. Mass spectra of 2-phenylbenzofuran, Figure S4. Proton NMR spectra of 2-phenylbenzofuran, Table S1. ICP data of initial K doped PdCu@C nanocatalyst, Table S2. ICP data of 5 time re-used K doped PdCu@ nanocatalyst.

**Author Contributions:** K.H.P. and S.P. provided academic direction, conceptualization, validation, funding acquisition, writing-reviewing and editing. S.J. and D.A. contributed equally on that work. S.J. contributed especially in the synthesis and analysis of materials, wrote the introduction, results and discussion, and conclusion parts. D.A. contributed regarding organic reactions and analysis of products, wrote the experimental part, checked English style as well as grammatical errors throughout the manuscript. S.S. and J.-S.B. contributed regarding the analysis of materials using XPS. All authors have read and agreed to the published version of the manuscript.

**Funding:** This research was supported by Basic Science Research Program through the National Research Foundation of Korea (NRF) grant funded by the Korea Government (MSIP) (NRF-2020R1I1A3067208 and NRF-2018R1D1A1B07045663).

**Data Availability Statement:** Data are contained within the article.

**Acknowledgments:** This research was supported by Basic Science Research Program through the National Research Foundation of Korea (NRF) grant funded by the Korea Government (NRF-2018R1D1A1B07045663, NRF-2020K1A3A7A09077715, NRF-2021M3H4A6A02045432, and NRF-2021M3I3A1084719) and by Korea Basic Science Institute (National Research Facilities and Equipment Center) grant funded by the Ministry of Education (grant No. 2021R1A6C101A429).

**Conflicts of Interest:** The authors declare no conflict of interest.

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
