# Peer review of "Non-Solvent Synthesis of a Robust Potassium-Doped PdCu-Pd-Cu@C Nanocatalyst for High Selectively Tandem Reactions"

_catalysts, doi:10.3390/catal11101191_

Round 1

Reviewer 1 Report

Park et al. prepared new K-doped PdCu@C by mechanically grinding and heating. Interesting and accurate appears the characterization of the catalyst that was tested in the tandem reaction of 2-benzofuran. The catalyst was resulted more active than other (commercial or prepared). Also the reuse of the catalyst was showed. I suggest accepting the manuscript after a careful check of the language in the text and in the abstract and after minor revisions:

  • proton NMR ??? I’d favour 1H NMR
  • 3 line 113-114: there is a lot of confusion between the figures
  • 8 line 243: Solvent for DMSO
  • 9 line 272: Magnesium was written with capital letter
  • 9 line 282: there is min-1, without superscript
  • 10 line 314: in the formula of magnesium sulfate, the number is not subscript

Author Response

Reviewer 1

Park et al. prepared new K-doped PdCu@C by mechanically grinding and heating. Interesting and accurate appears the characterization of the catalyst that was tested in the tandem reaction of 2-benzofuran. The catalyst was resulted more active than other (commercial or prepared). Also the reuse of the catalyst was showed. I suggest accepting the manuscript after a careful check of the language in the text and in the abstract and after minor revisions

Comment 1: proton NMR ??? I’d favour 1H NMR.

Response: We fully agree with the reviewer’s comment and opinion. We have re-drafted the manuscript provided. (Please see: Manuscript Page 6, Line 196-197)

Comment 2: 3 line 113-114: there is a lot of confusion between the figures.

Response: Thank you for your kind notice. We have re-drafted the manuscript provided. (Please see: Manuscript Page 3, Line 114)

Comment 3: 8 line 243: Solvent for DMSO.

Response: Thank you for your kind notice. We have re-drafted the manuscript provided (Please see: Manuscript Page 7, Line 243).

Comment 4: 9 line 272: Magnesium was written with capital letter.

Response: Thank you for your kind notice. We have re-drafted the manuscript provided (Please see: Manuscript Page 8, Line 272).

Comment 5: 9 line 282: there is min-1, without superscript.

Response: We sincerely appreciate reviewer’s kind. We have re-drafted the manuscript provided (Please see: Manuscript Page 8, Line 282).

Comment 6: 10 line 314: in the formula of magnesium sulfate, the number is not subscript.

Response: We sincerely appreciate reviewer’s kind and instructive comments on our work. We have added the reference provided (Please see: Manuscript Page 9, Line 314).

Reviewer 2 Report

The present paper describes a non-solvent synthesis of potassium-doped nanocatalysts efficiently used in tandem reactions of 2-phenylbenzofurans. The manuscript has been carefully prepared in terms of its content. I can recommend this article for publication in Catalysts after a little editorial correction:

1) Line 44 - 'physico-chemical' instead of 'physio-chemical'

2) Line 74 - 'effect' - please check the font

3) Line 235 - 'selective' instead of 'selectivity'

4) Line 314 - 'NaHCO3' instead of 'Na2HCO3'

5) Line 314 - 'MgSO4' instead of 'MgSO4'

6) Lines 446-447 - use italic for journal name abbreviation and volume 

                              number; use bold for year

7) Line 462 - use italic for journal abbreviation.

Author Response

The present paper describes a non-solvent synthesis of potassium-doped nanocatalysts efficiently used in tandem reactions of 2-phenylbenzofurans. The manuscript has been carefully prepared in terms of its content. I can recommend this article for publication in Catalysts after a little editorial correction

Comment 1: Line 44 - 'physico-chemical' instead of 'physio-chemical'.

Response: We fully agree with the reviewer’s comment and apologize the careless word. We have re-drafted the manuscript provided. (Please see: Manuscript Page 1, Line 44)

Comment 2: Line 74 - 'effect' - please check the font.

Response: Thank you for your kind notice. We have re-drafted the manuscript provided. (Please see: Manuscript Page 2, Line 74)

Comment 3: Line 235 - 'selective' instead of 'selectivity.

Response: We sincerely apologize for the grammatical mistake. We have re-drafted the manuscript provided. (Please see: Manuscript Page 7, Line 235).

Comment 4: Line 314 - 'NaHCO3' instead of 'Na2HCO3'.

Response: We sincerely appreciate reviewer’s kind. We have re-drafted the manuscript provided (Please see: Manuscript Page 9, Line 314).

Comment 5: Line 314 - 'MgSO4' instead of 'MgSO4'

Response: Thank you for your kind notice. We have re-drafted the manuscript provided (Please see: Manuscript Page 9, Line 314).

Comment 6: Lines 446-447 - use italic for journal name abbreviation and volume number; use bold for year

Response: We sincerely apologize for the template mistake. We have re-drafted the manuscript provided. (Please see: Manuscript Page 10, Line 446-447).

Comment 7: Line 462 - use italic for journal abbreviation.

Response: We sincerely apologize for the template mistake. We have re-drafted the manuscript provided. (Please see: Manuscript Page 10, Line 462).